1

2

3

# IMAGING SEISMIC WAVE-FIELDS WITH ALPARRAY AND NEIGHBORING EUROPEAN NETWORKS

Marcel Tesch<sup>1</sup>, Johannes Stampa<sup>1</sup>, Thomas Meier<sup>1</sup>, Edi Kissling<sup>2</sup>,

György Hetényi<sup>3</sup>, Wolfgang Friederich<sup>4</sup>, Michael Weber<sup>5</sup><sup>6</sup>, Benjamin Heit<sup>5</sup>,

and the AlpArray Working Group<sup>7</sup>

<sup>1</sup> Christian-Albrechts-Universität zu Kiel

<sup>2</sup> Eidgenössische Technische Hochschule Zürich

<sup>3</sup> Université de Lausanne

<sup>4</sup> Ruhr-Universität Bochum

 $^5$  Deutsches GeoForschungsZentrum Potsdam

<sup>6</sup> Universität Potsdam

<sup>7</sup> http://alparray.ethz.ch/

# Abstract

The modern-day coverage and availability of broad-band stations in the greater Alpine area offered by AlpArray,
 Swath-D and the European seismological networks allows for imaging seismic wave-fields at yet unprecedented
 resolution. In the AlpArray area and in Italy, the distance of any point to the nearest station is less than 30km,
 resulting in an average inter-station distance of about 45km. With a much denser deployment in a smaller region

<sup>9</sup> of the Alps (320km in length and 140km wide), the Swath-D network possesses an average inter-station distance <sup>10</sup> of about 15km.

<sup>11</sup> We provide single event seismogram sections, time slices of teleseismic and regional wave-fields, and wave-field <sup>12</sup> animations to reveal both the resolution capabilities of this dense station distribution as well as the enormous <sup>13</sup> spatio-temporal complexity of seismic wave propagation. The time slices and wave-field animations demonstrate <sup>14</sup> the need for dense regional arrays of broad-band stations, such as provided by AlpArray and neighboring networks, <sup>15</sup> to resolve properties of teleseismic wave-fields. Here we present the images of coherent arrivals of direct body and

<sup>16</sup> surface waves, multiple body wave reflections, and multi-orbit phases for teleseismic and regional events with

<sup>17</sup> moment magnitudes larger than 6 over a time window of at least 2:45 hours.

<sup>18</sup> Spatial observations of the wave-fields illustrate e.g. the decrease in horizontal wavelength from P to S to surface <sup>19</sup> waves and the way in which they considerably deviate from plane waves, due to heterogeneous earth structures

<sup>20</sup> along the path from the source to the array and beneath the regional array itself. Tomographic imaging techniques

for the deep structure beneath the regional array have to take this spatio-temporal variability into account and correct for it.

The lateral resolution of the regional broad-band array is however dependent on station density, in this case limited to about 100 km. Only even denser station distributions like those provided by Swath-D suffice to recover

<sup>25</sup> wave-fields of short period body and surface waves.

26

# INTRODUCTION

Already in 1889 Rebeur-Paschwitz suggested in the first description of the recording of a teleseismic event (at seismometers in Hamburg and Potsdam) to build a global network of identical stations to monitor the world-wide seismicity (Rebeur-Paschwitz, 1889). The ca. 100 Wiechert-seismometers then deployed world-wide until the

<sup>30</sup> 1920's were part of such a network, but it was not until the 1960's that the ca. 120 stations of the *World-Wide* 

<sup>31</sup> Standardized Seismograph Network (WWSSN) created the global infrastructure needed, including the data-exchange

<sup>32</sup> procedures and station technical capabilities (e.g. Oliver and Murphy, 1971). This allowed for the first time to

<sup>33</sup> construct seismogram sections for the whole earth (Müller and Kind, 1976).

Fig. 1: All available European broadband stations for event No. 1, Taiwan (C201802061550A). See Tab. 1 for additional information. Triangles mark individual stations, reference stations for Fig. 2 are prominently marked and labeled. Temporary AlpArray component (Z3) in blue, Swath-D (ZS) in red, LOBSTER as white circles, all other networks in gray.