# Peer review of "IMAGING SEISMIC WAVE-FIELDS WITH ALPARRAY AND NEIGHBORING EUROPEAN NETWORKS"

_Solid Earth, 2020_

## Referee Comment (RC1) · Anonymous Referee #1 · 17 Sep 2020

The authors proceed to an interesting and rather recent exercise in the field of passive seismology, which consists in processing the seismological records of natural earthquakes at dense large-aperture arrays and visualize ground motion vibrations as a function of space and time in a map (see e.g. IRIS Data Products http://ds.iris.edu/ds/products/gmv/).

As pointed out in the conclusion section of the article, this exercise is interesting for reaching a broad audience public, and there is probably also a potential for using such animations in joint science and graphic art experiments.

However, my first major comment is to question the general relevance of the exercise in the broad context of Geophysics: * What do we learn about the Earth? * What do we learn about the physics of wave-propagation? * Apart the images and videos, what
are the results/products of this analysis, and how and why other scientists should use these? This information is not missing but spread over small portions of the manuscript, in result sections associated to each earthquake. Consequently the reader does not immediately grasp the full utility of such an approach. I would recommend better introducing the approach in introduction, with its impacts and benefits.

My second major comment is about the displayed goal of revealing the resolution capabilities of the network and the spatial complexity of the wavefield through seismogram sections, times slices and wavefield animations. Are these standard ways of judging the resolution capabilities of an array? What about other tools classically used in array seismology (array response functions, beamforming, importance for eikonal tomography or time reversal imaging, in tomography)? Such a goal does not seem to be completely fulfilled because the analysis is mostly qualitative, and would require being more quantitative. The authors suggest that the wavefield spatio-temporal complexity should be accounted for in tomography, but a question is what is the amount of the deviation with respect to an unperturbed case, and is this amount so large that tomography, with its parameterization, damping, or smoothing, makes a large mistake by taking wave propagation in 1D Earth's reference models? Could a quantitative estimate of the spatio-temporal complexity come from the extraction of first-order perturbations (scattered wavefield) with respect to a reference state (direct wavefield)? The records could for instance be processed with a principal component analysis to separate both components. Another mean of judging the capability of the Array could be comparing the wavefield reconstruction with and without AlpArray.

Finally, my third major comment is about the organization of the paper. The Discussion section is a mixture between a result section for each individual earthquake, and in each of these result sections, a discussion of the significance of the results. The article would benefit in presenting instead the results by type of observations of interest, e.g. 1. Spotting polarity and timing errors; 2. Wavelength of the wavefield; 3. Examples of scattering and resolution. 4. Spotting exotic phases. 5. Dispersion anomalies.

6. Amplitude anomalies; and then discuss the significance for the physics of wave propagation.

Minor comments:

Page 1 line 31, left: Deviation from a plane wave. Any quantitative estimate? Page 1 line 25, right: Seismic arrays: why not using classical tools from array seismology? Page 1 line 35, left: I would say that the goal of tomography is to use (not really correct) this spatio temporal variability to image the structure at the origin of the variability. Page 3 line 26, right: Why these 6 events and not others? Why are they representative? Page 3 line 40, left: Maybe compare with Hi-Net / F-Net? Page 3 line 55, left: Is Faccenna et al. (2001) the first reference to introduce the concept of nappe stacking? This is ambiguous. Page 3 line 37, right: how to go beyond the "illustration" of spatial resolution capabilities? Page 4, line 13, right: I am quite surprised that the large amplitude surface-wave past 45 s appears as small oscillations in your gray-scale background plot when the earlier long period wavetrain appears a lot stronger in the image. Is the result of amplitude normalization by the envelope, or an effect of time sampling (binning)? Page 4, line 8, left, and Page 5, line 17, left: Concerning the processing, it is said that "all traces are detrended, instrument response-corrected, band-pass filtered between 100-500 s, and resampled to 1Hz" (page 4 line 8, left). This is long-period. Ok for looking at surface waves but what about body (P, S) waves? Figures 16 & 17 show ∼3 cycles of a P-wave in a 1 min window, so I suppose that this particular figure does not use the same bandpass as in the basic processing. Page 5, line 48, left: 8S, 9S, [...] →1. These are faint. 2. How can you be sure that these are corresponding phases? I think a close-up is needed for better demonstration... Page 5, line 47, right: Discussion → Is this a discussion or a result section? In a discussion, instead of focusing on events themselves, maybe you could focus on the particular features that you recognized in this data (see my major comment #3). Page 6, line 7, left: "artifact of projection" but what is the projection used here? Page 7, line 7, left: Aren't body-waves little dispersive? What difference in frequency content is there between the early and

late part of the P-wave coda? Why do long-periods arrive late? Maybe a spectrogram could help? Which mechanism do you think is at the origin of this dispersion? Page 8, line 8, right: Fig. 7 -> Fig. 6 Page 8, line 11, right: spot instrumental problems → This is an interesting application, that could be introduced / listed in introduction and possibly abstract. Page 9 line 28, left: Fig. 8-12 → why this reference to 4 figures when considering only the P/Pdiff case? Page 9 line 45, right: Is this dispersion opposite to the one observed on R1? If so why? Page 15 line 14-16, left: "Furthermore, amplifications in narrow bands often oriented almost parallel to the propagation direction are frequently observed". Could you show examples of these?

---

## Referee Comment (RC2) · Petr Kolínsky (Referee) · 28 Oct 2020

note A: I have made my opinion on the proposed manuscript before the other (anonymous) review was published on SED online discussion web page. Some of my suggestions may be close to those mentioned in the other review, however, my review was written independently.

note B: Two of the papers where I, P.K., am the first author, were already used in the original manuscript. I am using these references throughout my review. These are then not new references which I would ask the authors to add to the manuscript.

General remarks The paper uses a data from hundreds (up to 1600) of stations over the entire Europe. Big portion of the data comes from the AlpArray experiment comple-

mented also by the denser Swath D project. The data are uniformly processed, records are shown, animations presented. The work systematically browse the records in time and space commenting on many interesting features of wave propagation and presenting original way of plotting the seismogram section using binning of amplitudes in space and time. Showing and discussing the core of the seismological observation – the earthquake records – deserves an attention and should be acknowledged. The paper describes fundamental seismic phenomena, some of them visible and observable only thanks to the networks covering large region as well as thanks to the dense station coverage of the region at the same time. As it is one of up to now only a few papers exploiting AlpArray data in its entirety and showing the capability of such a project to advance or knowledge about the wave propagation, it should deserve an attention. Everything what has been done in the paper is concise, well described, documented in figures. However, my impression is, that the work ends up at half of the way. After such a beautiful observation, one would expect quantitative measurement of at least some of the phenomena mentioned. Even without modeling and inversion for structure, the effects – observed and commented based on visual inspection – could be maybe quantified and compared with papers observing similar distortion of the wavefield, both in case of body wave as well as surface waves. Below, I am giving more focused suggestions and comments.

Abstract The abstract promises a lot and after reading it, one gets easily excited and motivated to read the whole paper. However, the expectations are then not fully confirmed later.

We have revised the abstract to better reflect the goals of this paper.

Introduction The Introduction starts with historical overview, it gives a broad point of view but it is still reasonably short, showing the most important steps in building networks. As it is all about Europe, I would maybe even mention the PASSEQ project (2006 – 2008). The end of the Introduction already bit lowers the expectations talking about "imaging of the propagation". Which the paper is about, true. But after "imaging", one would expect some quantitative measurement, hypothesis or explanation.

Events & Data – Seismogram section Single-station approach allows for the measurement being applied independently of the network density. This is good in principle – you can obtain the same measurement both inside and outside of the dense AlpArray (Swath-D) networks. All processing steps are clearly explained and the record sections and animations are based on properly documented data processing. All the comments of observed phenomena, however, are based on the visual impression of the wavefield propagation over the dense networks. My main point is: is there is a way how to take advantage of the dense networks (let's say inside the AlpArray region at least) to map the wavefronts in time and space with some reasonable smoothing among the stations? How to quantify the distortions of the wavefield?

These are all very relevant questions which are in part the topic of ongoing research by the authors, the results of which will be presented in separate future publications.

Discussion This section should be more properly called "Observation" or maybe "Measurement". It describes qualitatively the observation. It uses a lot terms like: "varies considerably", "mostly aligned", "slight deformations", "visibly deformed"; "heavily interfering", "severely deformed", "significant deformation", "phase bundles", "unaffected by distortions", "rather coherent", "seems aligned", "notably sensitive", "perceptibly different". This would be alright, if it was supplemented by numbers and plots showing the distortions. Could it be possible to somehow track the wavefronts for distinct phases, maybe similarly like in Kolínsky et al. (2020), see Fig. C1 – bottom panel, where the zero-crossings of surface waves at given period are plotted and smoothed over the map. This is also purely one-station measurement. Or, at the top panel of the same figure, the group maxima are plotted, again a one-station measurement (see also animations in the online-only Supplement to the latter paper).

Here, I list some questions, which can be maybe answered or at least documented if a quantitative measurement is applied on the wavefield:

- Horizontal wavelength of body waves. You give rough estimates for some of the earthquakes of the apparent wavelength of the body waves striking the network. Can you evaluate the connection with the incidence angle? Does the apparent horizontal wavelength of later arrival as PP correspond to the geometry of P and PP incidence angles

(for different epicentral distances)? How does this trade off with frequency? Apparent horizontal wavelength could be caused by different frequency as well as different incidence angle, or both.

- Diffraction of body waves. If you quantify the distorted wavefronts, what are the time delays by which the real wavefronts deviate from the circular 1D wavefronts? Do you see some systematic behavior, connected to frequency, incidence angle, number of bounces (P, PP, PPP, ...), epicentral distance? What about comparison with the papers using the body wave diffraction like Cottaar and Romanowicz (2012) and Yuan and Romanowicz (2017) (both using S/Sdiff phases, though)?

- I see two options, how to visualize the wavefronts of various phases. You can plot the picks in the sections of records directly, what would, however, miss the azimuthal dependence. But it would allow you to plot several (all) phases into one record section. Or you plot the picks in the maps as a contours of arrival time. This would allow to see the spatial distribution (and distortion), but each phase would need a separate figure.

- Surface waves: Can you plot some of the standard deviation maps only for the surface-wave time window? Or better even for filtered narrow-band of frequencies? Does it show the stripes as expected (and also shown) by the studies from USArray and AlpArray? Can you then say, to which extend the amplitude variations come from the local structure and which portion must have been brought to the region from outside thanks to the propagation complexities? Are there similarities between the earthquakes?

- You comment both on the amplitudes and distortion of the wavefronts. These are probably mutually connected. Focusing and defocusing takes place whenever the wavefront is curved. Can you compare the two observations?

Minor comments technical: - The animations are amazing. The files are, however, bit too huge. It is difficult to download them and even difficult to playback them on a reasonable computer as the memory demands are high. Also, the background topography map includes unnecessary details (contours), which can be removed to make the image less busy.

- The reference to EGU abstract "Kolínsky, P., Bokelmann, G., AlpArray Working Group, Upper Mantle Imaging with AlpArray Surface Wave Diffraction: The Cameroon Volcanic Line, Geophysical Research Abstracts, Vol. 21, 2019" can be replaced by the paper "Kolínsky, P., F. M. Schneider & G. Bokelmann, 2020. Surface wave diffraction pattern recorded on AlpArray: Cameroon Volcanic Line case study, J. Geophys. Res: Solid Earth, 125, e2019JB019102, doi: 10.1029/2019JB019102.". It would also help to avoid two references with the same citation in the text "Kolínsky et al., 2019". Moreover, the citation in the text is used only once and hence one of the references in the reference list is probably not needed.

comments line by line: line 102 "They ..." –> "The records ..."

line 103 "half a sample width due to station effects" –> "half the sampling interval due to different digitizer time stamping"

lines 101-104: This sounds like if the records were FIRST resampled to 1sps and THEN rounded to the nearest second. It would be better to first align the seconds in the original sampling and then resample it as the error would be much smaller.

lines 122 – 131 Here it is not clear to me, how the labels of SS, SSS, 4S and so on made it into the plot. You talk about observation. However, when removing the 4S to 9S labels, it is difficult to see six different phases coming in the given time window. How did you "observe" these six distinct phases? Or are the labels marked at times predicted by some 1D model? There are clearly coherent arrivals, I just wonder, how can you distinguish between them. The caption to Fig. 2 says: "Corresponding theoretical travel-times were computed with TauP". But where do we see these theoretical travel times? Should it be the position and size of the labels? I would also add the sentence about the theoretical times into the main text.

line 139: This follows my previous point. Visually, I can see the phases, but to say, you have "detected" them, I would expect some quantitative analysis, some picks, or maybe a line drawn over the plot in Fig. 2 to show how can we really separate 8S from 9S.

A general remark to the whole Section "Seismogram Section"> It is all about the vertical component, right? Wouldn't all the S-phases be better visible on the transverse component?

Fig. 2> The red/blue color behind the selected traces probably shows the envelopes, the same as in the animations (reference trace)? If so, please, say it in the caption or in the text.

line 159> "to indicating" –> "indicating"

line 163> "Discussion". As already mentioned above, I would call this section probably "Observation".

line 170> "small circles" –> "theoretical wavefronts"; even "small circles" is not wrong, here it is bit misleading and could make a confusion with all the small circles showing the amplitude of ground velocity

lines 170-174> While the first point about the theoretical wavefronts being apparently curved in "wrong" direction could be commented as "an artifact of the map projection", the second, however, is not a problem of the projection, but simply reflecting the fact, that the waves really do arrive from northeast, as it is the shortest way to the epicenter. So, in fact, the source IS located to the northeast. When sitting in the Alps and pointing your finger to Taiwan, you will point to northeast.

line 180> "dispersion of body waves"; here I would add a reference or some more explanation about this effect. This is actually very interesting topic, which could be reflected also later in the discussion.

lines 181-190> Here one tackles the main issue, that the wavefronts of the wavefield

are compared with the theoretical wavefronts. However, as opposed to the theoretical ones, which are drawn by solid lines, the observed wavefronts are not drawn, one can only visually guess them. Here I would expect an animated "measured" wavefront overlaid on top of the figure and based on connecting the same phase (maximum/minimum/zero crossings) over the network, maybe similarly as in the animations in the Supplement to paper by Kolínsky et al. (2020), see above, where this is done for zero-crossings of the Rayleigh waves.

line 198> You say, that "They cannot be properly imaged by the available station density." (meaning waves shorter than 20 s), but at line 145 you said, you have anyway removed all waves shorter than 20 s.

line 201> "previously discussed event" – This is the first event described here, so it is not clear which is the previous one.

line 207> What could be the precision of the timing errors you are able to spot here? Units of seconds? 1 sec? 5 sec?

line 209+210> Yes, I do see the outliers. It would be nice, if directly the (five?) station names were given here.

Fig. 7> If I got it right, the color shows the standard deviation, and the size of the circles shows the amplitude? Which is somehow the same information here? Is this why the blue circles are small and the red circles are big? If the color and size really show the same, I would keep the color scale but I would draw all the circles by the same size, so that the red does not dominate over the blue. Or is the size of the circles showing the instantaneous amplitude of the moment of the snapshot and the color the deviation over the 2:45 hour? Please, specify it in the figure caption. However, still, the correlation between the size and the color is striking.

lines 216-221> The deviations are given for a very long time window of 2:45 hrs. All the different arrivals of body waves as well as surface waves and coda are included in

the deviations. I think, that the discussion about if the deviations indicate the scattering outside of the array or some heterogeneities beneath the array would be better constrained, if the deviations are split by the wave types, meaning, if they are calculated and discussed by smaller time windows.

line 229> "small-scale local seismic activity" – This sounds really interesting, do you have a suspicion on some local earthquake or quarry blast to produce these signal? Or what could it be, that "small-scale local seismic activity"?

lines 271 – 276> This is really interesting. Can you tell, if the effect of non-decreasing amplitudes with distance is rather connected to the local structure or to the propagation effects? You speculate about both. What about comparison with other earthquakes? Fig. 13 may look like if there were stripes of high and low amplitudes, similarly as in Kolínsky and Bokelmann (2019). As the time window is, however, that long, it is difficult to say it clearly. What if you plot the same only for the time window when R1 arrives?

line 332 and Fig. 21> Here come the clear stripes! Exciting! Let's discuss it more also with respect to Fig. 13. You have one sentence here, but this is a topic to spent bit more text on it. (Here the reference K+B(2019) is ambiguous, probably pointing to the paper in GJI? {and not the EGU abstract?}, see my comment above.)

line 361+362> Yes, this is the main point. How to extract local structure from wavefields distorted by propagation effects.

line 366> "frequently observed" ... I would add the references here, namely the USArray papers by Pollitz (2008), Liang and Langston (2009), Lin et al. (2012), Foster et al. (2014), Liu and Holt (2015), and Chen et al. (2018) from NECESSArray, and maybe repeat also Kolínsky and Bokelmann (2019) for AlpArray.

To conclude, I am considering the proposed observation to be of high interest. To fully exploit the data and to make clearer conclusion or at least hypothesis about the origin of the distortions, I would suggest to complement the study with quantitative analysis

of the observation. I suggest the manuscript goes for major revision.

references used in this review:

Pollitz, F.F., 2008. Observations and interpretation of fundamental mode Rayleigh wavefields recorded by the Transportable Array (USArray), J. Geophys. Res., 113, doi:10.1029/2007JB005556.

Liang, Ch. & Ch. A. Langston, 2009. Wave gradiometry for USArray: Rayleigh waves, J. Geophys. Res., 114, B02308, doi:10.1029/2008JB005918.

Lin, F.-C., Tsai, V.C. & Ritzwoller, M.H., 2012. The local amplification of surface waves: A new observable to constrain elastic velocities, density, and anelastic attenuation, J. Geophys. Res., 117, B06302, doi:10.1029/2012JB009208.

Foster, A., G. Ekström, & V. Hjörleifsdóttir, 2014. Arrival-angle anomalies across the USArray Transportable Array, Earth and Planetary Science Letters, 402, 58-68, doi:10.1016/j.epsl.2013.12.046.

Liu, Y. & W. E. Holt, 2015. Wave gradiometry and its link with Helmholtz equation solutions applied to USArray in the eastern U.S., J. Geophys. Res., 120, 5717-5746, doi:10.1002/2015JB011982.

Chen, H., S. Ni, R. Chu, J. Chong, Z. Liu & Zhu, L., 2018. Influence of the off-great-circle propagation of Rayleigh waves on event-based surface wave tomography in Northeast China, Geophys. J. Int., 214, 1105–815 1124.

Kolínsky, P., G. Bokelmann & the AlpArray Working Group, 2019. Arrival angles of teleseismic fundamental mode Rayleigh waves across the AlpArray, Geophys. J. Int., 218, 114-144, doi: 10.1093/gji/ggz081.

Kolínsky, P., F. M. Schneider & G. Bokelmann, 2020. Surface wave diffraction pattern recorded on AlpArray: Cameroon Volcanic Line case study, J. Geophys. Res: Solid Earth, 125, e2019JB019102, doi: 10.1029/2019JB019102.

Cottaar, S. & B. Romanowicz, 2012. An unusually large ULVZ at the base of the mantle near Hawaii, Earth and Planetary Science Letters, 355-356, 213-222, doi: 10.1016/j.epsl.2012.09.005.

Yuan, Y. & B. Romanowicz, 2017. Seismic evidence for partial melting at the root of major hot spot plumes, Science, 357, 393-397, doi:10.1126/science.aan0760.

————————————————————

---

## Author Comment (AC1) · 21 Dec 2020

We thank the reviewer for their thorough and constructive feedback!

The authors proceed to an interesting and rather recent exercise in the field of passive seismology, which consists in processing the seismological records of natural earthquakes at dense large-aperture arrays and visualize ground motion vibrations as a function of space and time in a map (see e.g. IRIS Data Products http://ds.iris.edu/ds/products/gmv/).

As pointed out in the conclusion section of the article, this exercise is interesting for reaching a broad audience public, and there is probably also a potential for using such animations in joint science and graphic art experiments.

However, my first major comment is to question the general relevance of the exercise in the broad context of Geophysics: * What do we learn about the Earth? * What do we learn about the physics of wave-propagation? * Apart the images and videos, what are the results/products of this analysis, and how and why other scientists should use these? This information is not missing but spread over small portions of the manuscript, in result sections associated to each earthquake. Consequently the reader does not immediately grasp the full utility of such an approach. I would recommend better introducing the approach in introduction, with its impacts and benefits.

We revised the abstract and introduction to more succinctly state the utility provided by spatial observations of this nature. Examples include the better understanding of properties of the wave-field and its individual phases, or the necessity for correction of deformations for structural analysis. The use as a tool for data quality assessments and identification of faulty stations is also relevant.

My second major comment is about the displayed goal of revealing the resolution capabilities of the network and the spatial complexity of the wavefield through seismogram sections, times slices and wavefield animations. Are these standard ways of judging the resolution capabilities of an array?

The point is, that this is the first time in history where a seismological array provides the ability to spatially resolve wavefields from *single-event datasets* at such scale. Similar seismogram sections have been obtained previously by stacking waveforms of many events. In that sense it is not a standard way of judging resolution capability, as it wasn't possible before. We revised the manuscript to better reflect this.

What about other tools classically used in array seismology (array response functions, beamforming, importance for eikonal tomography or time reversal imaging, in tomography)? Such a goal does not seem to be completely fulfilled because the analysis is mostly qualitative, and would require being more quantitative.

Absolutely. The assessment of these datasets ultimately serves the purpose of understanding the kinds of quantitative methods that are now being enabled by them. Each of the mentioned examples would however require their own stand-alone publication to discuss them in the context of the AlpArray, SwathD, and European Networks, which therefore puts them thoroughly outside the scope of this paper. We more clearly expressed this sentiment in the Introduction to delineate our goals.

The authors suggest that the wavefield spatio-temporal complexity should be accounted for in tomography, but a question is what is the amount of the deviation with respect to an unperturbed case, and is this amount so large that tomography, with its parameterization, damping, or smoothing, makes a large mistake by taking wave propagation in 1D Earth's reference models? Could a quantitative estimate of the spatio-temporal complexity come from the extraction of first-order perturbations (scattered wavefield) with respect to a reference state (direct wavefield)? The records could for instance be processed with a principal component analysis to separate both components.

The reviewer poses a few questions that we are trying to answer with our ongoing research which, again, is best served by another separate publication. By visualizing the data in the spatial-temporal domain as done in this paper, we can clearly tell that the available resolution now offers a path to account for wavefield complexity in tomographic inversions if dense stations configurations are available. This is now clarified in the text.

Another mean of judging the capability of the Array could be comparing the wavefield reconstruction with and without AlpArray.

To make this comparison we deliberately included all available stations surrounding the Alpine region. We believe the difference in wavefield reconstruction inside vs. outside of AlpArray is quite striking in our figures. We now point that out more clearly in the text.

Finally, my third major comment is about the organization of the paper. The Discussion

section is a mixture between a result section for each individual earthquake, and in each of these result sections, a discussion of the significance of the results. The article would benefit in presenting instead the results by type of observations of interest, e.g. 1. Spotting polarity and timing errors; 2. Wavelength of the wavefield; 3. Examples of scattering and resolution. 4. Spotting exotic phases. 5. Dispersion anomalies. 6. Amplitude anomalies; and then discuss the significance for the physics of wave propagation.

We believe it is important for the paper to discuss the animations event-by-event, as they are the main product of this publication. We want the reader to be able to step through each animation and have a guideline at hand that points out and describes its features at any given time. To break these observations up as suggested would not serve that purpose. We have however rewritten the Abstract, Introduction, and Conclusions to better point out the observations of interest and to summarize the significant results.

Minor comments: Page 1 line 31, left: Deviation from a plane wave. Any quantitative estimate?

(see above)

Page 1 line 25, right: Seismic arrays: why not using classical tools from array seismology?

Classical array processing techniques based on the assumption of plane waves are not appropriate to analyse the wavefield complexities observed by dense regional arrays. This is now more clearly emphasized in the text.

Page 1 line 35, left: I would say that the goal of tomography is to use (not really correct) this spatio temporal variability to image the structure at the origin of the variability.

Corrected.

Page 3 line 26, right: Why these 6 events and not others? Why are they representative?

The events cover a range of azimuths and distances (both local and teleseismic) and feature similar magnitudes, so that they can be compared within reason. After assessing a number of events we decided that this set best represented the properties we wanted to highlight. The text has been amended to make that clearer.

Page 3 line 40, left: Maybe compare with Hi-Net / F-Net?

These networks are now mentioned in the text.

Page 3 line 55, left: Is Faccenna et al. (2001) the first reference to introduce the concept of nappe stacking? This is ambiguous.

Further references have been added and the text is now formulated more clearly.

Page 3 line 37, right: how to go beyond the "illustration" of spatial resolution capabilities?

(see above)

Page 4, line 13, right: I am quite surprised that the large amplitude surface-wave past 45 s appears as small oscillations in your gray-scale background plot when the earlier long period wavetrain appears a lot stronger in the image. Is the result of amplitude normalization by the envelope, or an effect of time sampling (binning)?

This is caused by the envelope weighting. The individual seismograms plotted on top have the same time sampling as the background for comparison. Binning has almost no influence, as the bins are so small that usually only a few traces are averaged per bin.

Page 4, line 8, left, and Page 5, line 17, left: Concerning the processing, it is said that "all traces are detrended, instrument response-corrected, band-pass filtered between 100-500 s, and resampled to 1Hz" (page 4 line 8, left). This is long-period. Ok for looking at surface waves but what about body (P, S) waves? Figures 16 17 show âĹij3 cycles of a P-wave in a 1 min window, so I suppose that this particular figure does not

use the same bandpass as in the basic processing.

Yes, the processing as described is applicable to the seismogram sections, animations, and time slices.

Page 5, line 48, left: 8S, 9S, [. . .] →1. These are faint. 2. How can you be sure that these are corresponding phases? I think a close-up is needed for better demonstration...

We computed synthetic arrival times for many higher order body wave phases to be able to properly identify them in the section, so though they are faint, they are discerable and they are exactly where we would expect them. In the text we also clarifiy why some of them are labelled twice, due to arrivals from opposite directions.

Page 5, line 47, right: Discussion → Is this a discussion or a result section? In a discussion, instead of fo- cusing on events themselves, maybe you could focus on the particular features that you recognized in this data (see my major comment 3).

As addressed for the previous comment, the discussion has been revised accordingly.

Page 6, line 7, left: "artifact of projection" but what is the projection used here?

It's an equirectangular projection. Text amended.

Page 7, line 7, left: Aren't body-waves little dispersive? What difference in frequency content is there between the early and late part of the P-wave coda? Why do long-periods arrive late? Maybe a spectrogram could help? Which mechanism do you think is at the origin of this dispersion?

Body waves are little dispersive but they ultimately *are* dispersive. Analysis of body wave dispersion would require again an entire publication on its own.

Page 8, line 8, right: Fig. 7 -> Fig. 6 Page 8, line 11, right: spot instrumental problems → This is an interesting application, that could be introduced / listed in introduction and possibly abstract.

Has been added.

Page 9 line 28, left: Fig. 8-12 → why this reference to 4 figures when considering only the P/Pdiff case?

That was a mistake, has been corrected.

Page 9 line 45, right: Is this dispersion opposite to the one observed on R1? If so why?

It is. This is a result of the dramatically increased damping introduced by an entire additional orbit as well as the increased separation of the frequencies at such distances. Text has been amended.

Page 15 line 14-16, left: "Furthermore, amplifications in narrow bands often oriented almost parallel to the propagation direction are frequently observed". Could you show examples of these?

An example is given in Fig. 21. This is now clarified in the text. References to publications with similar observations have been added.

---

## Author Comment (AC2) · 21 Dec 2020

We thank Petr for his detailed and constructive review of the manuscript!

note A: I have made my opinion on the proposed manuscript before the other (anonymous) review was published on SED online discussion web page. Some of my suggestions may be close to those mentioned in the other review, however, my review was written independently.

note B: Two of the papers where I, P.K., am the first author, were already used in the original manuscript. I am using these references throughout my review. These are then not new references which I would ask the authors to add to the manuscript.

General remarks The paper uses a data from hundreds (up to 1600) of stations over

the entire Europe. Big portion of the data comes from the AlpArray experiment complemented also by the denser Swath D project. The data are uniformly processed, records are shown, animations presented. The work systematically browse the records in time and space commenting on many interesting features of wave propagation and presenting original way of plotting the seismogram section using binning of amplitudes in space and time. Showing and discussing the core of the seismological observation – the earthquake records – deserves an attention and should be acknowledged. The paper describes fundamental seismic phenomena, some of them visible and observable only thanks to the networks covering large region as well as thanks to the dense station coverage of the region at the same time. As it is one of up to now only a few papers exploiting AlpArray data in its entirety and showing the capability of such a project to advance or knowledge about the wave propagation, it should deserve an attention. Everything what has been done in the paper is concise, well described, documented in figures. However, my impression is, that the work ends up at half of the way. After such a beautiful observation, one would expect quantitative measurement of at least some of the phenomena mentioned. Even without modeling and inversion for structure, the effects – observed and commented based on visual inspection – could be maybe quantified and compared with papers observing similar distortion of the wavefield, both in case of body wave as well as surface waves. Below, I am giving more focused suggestions and comments.

As mentioned in our response to Review No. 1, any kind of detailed quantitative analysis would require the room and focus of an entire dedicated publication on its own. It is well beyond the scope of this paper to quantify the observed wavefield deformations and it would not serve the overall purpose of illustrating seismological phenomena using wavefield animations.

Abstract The abstract promises a lot and after reading it, one gets easily excited and motivated to read the whole paper. However, the expectations are then not fully confirmed later.

[Figure]

We have revised the abstract to better reflect the goals of this paper.

Introduction The Introduction starts with historical overview, it gives a broad point of view but it is still reasonably short, showing the most important steps in building networks. As it is all about Europe, I would maybe even mention the PASSEQ project (2006 – 2008).

We have added PASSEQ to the list of networks as suggested.

The end of the Introduction already bit lowers the expectations talking about "imaging of the propagation". Which the paper is about, true. But after "imaging", one would expect some quantitative measurement, hypothesis or explanation.

(see above)

Events Data – Seismogram section Single-station approach allows for the measurement being applied independently of the network density. This is good in principle – you can obtain the same measurement both inside and outside of the dense AlpArray (Swath-D) networks. All processing steps are clearly explained and the record sections and animations are based on properly documented data processing. All the comments of observed phenomena, however, are based on the visual impression of the wavefield propagation over the dense networks. My main point is: is there is a way how to take advantage of the dense networks (let's say inside the AlpArray region at least) to map the wavefronts in time and space with some reasonable smoothing among the stations? How to quantify the distortions of the wavefield?

These are all very relevant questions which are in part the topic of ongoing research by the authors, the results of which will be presented in separate future publications.

Discussion This section should be more properly called "Observation" or maybe "Measurement". It describes qualitatively the observation. It uses a lot terms like: "varies considerably", "mostly aligned", "slight deformations", "visibly deformed"; "heavily interfering", "severely deformed", "significant deformation", "phase bundles", "unaffected by

distortions", "rather coherent", "seems aligned", "notably sensitive", "perceptibly differ-
ent". This would be alright, if it was supplemented by numbers and plots showing the
distortions. Could it be possible to somehow track the wavefronts for distinct phases,
maybe similarly like in Kolínsky et al. (2020), see Fig. C1 – bottom panel, where the
zero-crossings of surface waves at given period are plotted and smoothed over the
map. This is also purely one-station measurement. Or, at the top panel of the same
figure, the group maxima are plotted, again a one-station measurement (see also ani-
mations in the online-only Supplement to the latter paper).

Here, I list some questions, which can be maybe answered or at least documented if a
quantitative measurement is applied on the wavefield:

- Horizontal wavelength of body waves. You give rough estimates for some of the earth-
quakes of the apparent wavelength of the body waves striking the network. Can you
evaluate the connection with the incidence angle? Does the apparent horizontal wave-
length of later arrival as PP correspond to the geometry of P and PP incidence angles
(for different epicentral distances)? How does this trade off with frequency? Appar-
ent horizontal wavelength could be caused by different frequency as well as different
incidence angle, or both.

(see above)

- Diffraction of body waves. If you quantify the distorted wavefronts, what are the time
delays by which the real wavefronts deviate from the circular 1D wavefronts? Do you
see some systematic behavior, connected to frequency, incidence angle, number of
bounces (P, PP, PPP, ...), epicentral distance? What about comparison with the papers
using the body wave diffraction like Cottaar and Romanowicz (2012) and Yuan and
Romanowicz (2017) (both using S/Sdiff phases, though)?

(see above)

- I see two options, how to visualize the wavefronts of various phases. You can plot

Interactive
comment

the picks in the sections of records directly, what would, however, miss the azimuthal dependence. But it would allow you to plot several (all) phases into one record section. Or you plot the picks in the maps as a contours of arrival time. This would allow to see the spatial distribution (and distortion), but each phase would need a separate figure.

(see above)

- Surface waves: Can you plot some of the standard deviation maps only for the surface-wave time window? Or better even for filtered narrow-band of frequencies? Does it show the stripes as expected (and also shown) by the studies from USArray and AlpArray? Can you then say, to which extend the amplitude variations come from the local structure and which portion must have been brought to the region from outside thanks to the propagation complexities? Are there similarities between the earthquakes?

(see above)

- You comment both on the amplitudes and distortion of the wavefronts. These are probably mutually connected. Focusing and defocusing takes place whenever the wavefront is curved. Can you compare the two observations?

(see above)

Minor comments technical:

- The animations are amazing. The files are, however, bit too huge. It is difficult to download them and even difficult to playback them on a reasonable computer as the memory demands are high. Also, the background topography map includes unnecessary details (contours), which can be removed to make the image less busy.

We amended the supplementary materials with lower resolution versions of the animations.

- The reference to EGU abstract "Kolínsky, P., Bokelmann, G., AlpArray Working Group,

Upper Mantle Imaging with AlpArray Surface Wave Diffraction: The Cameroon Volcanic Line, Geophysical Research Abstracts, Vol. 21, 2019" can be replaced by the paper "Kolínsky, P., F. M. Schneider G. Bokelmann, 2020. Surface wave diffraction pattern recorded on AlpArray: Cameroon Volcanic Line case study, J. Geophys. Res: Solid Earth, 125, e2019JB019102, doi: 10.1029/2019JB019102.". It would also help to avoid two references with the same citation in the text "Kolínsky et al., 2019". Moreover, the citation in the text is used only once and hence one of the references in the reference list is probably not needed.

The reference has been corrected.

comments line by line: line 102 "They ..." –> "The records ..."

Changed.

line 103 "half a sample width due to station effects" –> "half the sampling interval due to different digitizer time stamping"

Changed.

lines 101-104: This sounds like if the records were FIRST resampled to 1sps and THEN rounded to the nearest second. It would be better to first align the seconds in the original sampling and then resample it as the error would be much smaller.

This is correct, but this spurious error will not affect the animations. Yes, for detailed quantitative analyses this should be corrected.

lines 122 – 131 Here it is not clear to me, how the labels of SS, SSS, 4S and so on made it into the plot. You talk about observation. However, when removing the 4S to 9S labels, it is difficult to see six different phases coming in the given time window. How did you "observe" these six distinct phases? Or are the labels marked at times predicted by some 1D model? There are clearly coherent arrivals, I just wonder, how can you distinguish between them. The caption to Fig. 2 says: "Corresponding theoretical traveltimes were computed with TauP". But where do we see these theoretical travel

times? Should it be the position and size of the labels? I would also add the sentence about the theoretical times into the main text.

Yes, the labels mark theoretical arrival times that we calculated for a number of body wave phases. We agree that it is probably not possibly to distinguish 6 discrete phases between 4S and 9S by eye, but it is clear that we can see some coherent arrivals in that region and we have confirmed while creating the figures that they do indeed match up with the theoretical arrival times. The labels are there to put them into context. The text has been amended accordingly.

line 139: This follows my previous point. Visually, I can see the phases, but to say, you have "detected" them, I would expect some quantitative analysis, some picks, or maybe a line drawn over the plot in Fig. 2 to show how can we really separate 8S from 9S.

We experimented with plotting the theoretical travel times as lines on top of the section but ultimately decided that this was visually too cluttered and the signals themselves became less visible. Furthermore, the seismogram section is not intended to state that we have detected these phases in any kind of formalized way but rather that based on comparable single-event datasets it is *possible* to do so (which is far from self-evident without looking at the data), as we think is unequivocally shown by Fig. 2. This has been made more clear in the text.

A general remark to the whole Section "Seismogram Section"> It is all about the vertical component, right? Wouldn't all the S-phases be better visible on the transverse component?

Yes, it is the vertical component. It's true that some phases might have benefited from being viewed on the transverse or radial components. We added a remark in the text.

Fig. 2> The red/blue color behind the selected traces probably shows the envelopes, the same as in the animations (reference trace)? If so, please, say it in the caption or

in the text.

Yes, that is true. It has been added to the caption.

line 159> "to indicating" –> "indicating"

Corrected.

line 163> "Discussion". As already mentioned above, I would call this section probably "Observation".

Renamed to "Discussion of the Animations".

line 170> "small circles" –> "theoretical wavefronts"; even "small circles" is not wrong, here it is bit misleading and could make a confusion with all the small circles showing the amplitude of ground velocity

Good point. Has been changed.

lines 170-174> While the first point about the theoretical wavefronts being apparently curved in "wrong" direction could be commented as "an artifact of the map projection", the second, however, is not a problem of the projection, but simply reflecting the fact, that the waves really do arrive from northeast, as it is the shortest way to the epicenter. So, in fact, the source IS located to the northeast. When sitting in the Alps and pointing your finger to Taiwan, you will point to northeast.

That is correct. We have rephrased the sentence to better express this.

line 180> "dispersion of body waves"; here I would add a reference or some more explanation about this effect. This is actually very interesting topic, which could be reflected also later in the discussion.

Further explanation and references have been added.

lines 181-190> Here one tackles the main issue, that the wavefronts of the wavefield are compared with the theoretical wavefronts. However, as opposed to the theoreti-

cal ones, which are drawn by solid lines, the observed wavefronts are not drawn, one can only visually guess them. Here I would expect an animated "measured" wavefront overlaid on top of the figure and based on connecting the same phase (maximum/minimum/zero crossings) over the network, maybe similarly as in the animations in the Supplement to paper by Kolínsky et al. (2020), see above, where this is done for zero-crossings of the Rayleigh waves.

(see above)

line 198> You say, that "They cannot be properly imaged by the available station density." (meaning waves shorter than 20 s), but at line 145 you said, you have anyway removed all waves shorter than 20 s.

They have been removed exactly *because* one cannot expect to properly image them with the available station density. Text has been amended.

line 201> "previously discussed event" – This is the first event described here, so it is not clear which is the previous one.

Clarified.

line 207> What could be the precision of the timing errors you are able to spot here? Units of seconds? 1 sec? 5 sec?

In our case the smallest possible timing error is about 2s, but this is purely a function of the chosen sampling rate for the animation. A remark has been added to the text. It is now pointed out in the text that dense arrays provide new options for quality control of the data.

line 209+210> Yes, I do see the outliers. It would be nice, if directly the (five?) station names were given here.

Examples have been added.

Fig. 7> If I got it right, the color shows the standard deviation, and the size of the

circles shows the amplitude? Which is somehow the same information here? Is this why the blue circles are small and the red circles are big? If the color and size really show the same, I would keep the color scale but I would draw all the circles by the same size, so that the red does not dominate over the blue. Or is the size of the circles showing the instantaneous amplitude of the moment of the snapshot and the color the deviation over the 2:45 hour? Please, specify it in the figure caption. However, still, the correlation between the size and the color is striking.

Color and size of the circles always encode the \*same\* information. Small blue circles indicate a low standard deviation, big red circles a large one. Again, to enforce visual consistency across figures. This has been made clearer in the text.

lines 216-221> The deviations are given for a very long time window of 2:45 hrs. All the different arrivals of body waves as well as surface waves and coda are included in the deviations. I think, that the discussion about if the deviations indicate the scattering outside of the array or some heterogeneities beneath the array would be better constrained, if the deviations are split by the wave types, meaning, if they are calculated and discussed by smaller time windows.

The deviations are always comparable if they include R1. Hence, they were calculated for the same length as the animations. An quantitative analysis of amplitude variations of different wave types is beyond the scope of the paper.

line 229> "small-scale local seismic activity" – This sounds really interesting, do you have a suspicion on some local earthquake or quarry blast to produce these signal? Or what could it be, that "small-scale local seismic activity"?

Yes, the dense networks can also be used to detect local seismicity including quarry blasts or induced events using various algorithms. This is now pointed out in the text.

lines 271 – 276> This is really interesting. Can you tell, if the effect of non-decreasing amplitudes with distance is rather connected to the local structure or to the propagation

effects? You speculate about both.

It has to be both. Both local structure and propagation have the potential to reshape amplitude distributions significantly.

What about comparison with other earthquakes? Fig. 13 may look like if there were stripes of high and low amplitudes, similarly as in Kolínsky and Bokelmann (2019). As the time window is, however, that long, it is difficult to say it clearly. What if you plot the same only for the time window when R1 arrives?

It doesn't much matter how wide the window is, as long as R1 is included.

line 332 and Fig. 21> Here come the clear stripes! Exciting! Let's discuss it more also with respect to Fig. 13. You have one sentence here, but this is a topic to spent bit more text on it. (Here the reference K+B(2019) is ambiguous, probably pointing to the paper in GJI? and not the EGU abstract?, see my comment above.)

Yes, indeed. We elaborated on the discussion of this observation.

line 361+362> Yes, this is the main point. How to extract local structure from wavefields distorted by propagation effects.

As mentioned in the text Helmholtz tomography is one of the options.

line 366> "frequently observed" ... I would add the references here, namely the USArray papers by Pollitz (2008), Liang and Langston (2009), Lin et al. (2012), Foster et al. (2014), Liu and Holt (2015), and Chen et al. (2018) from NECESSArray, and maybe repeat also Kolínsky and Bokelmann (2019) for AlpArray.

References added.

To conclude, I am considering the proposed observation to be of high interest. To fully exploit the data and to make clearer conclusion or at least hypothesis about the origin of the distortions, I would suggest to complement the study with quantitative analysis of the observation. I suggest the manuscript goes for major revision.

Thank you for your feedback! To reiterate, quantitative analysis of spatial wavefront distortions is an area of high interest deserving further research.